# Pirfenidone Alleviates Against Fine Particulate Matter-Induced Pulmonary Fibrosis Modulating via TGF-β1/TAK1/MKK3/p38 MAPK Signaling Pathway in Rats

**DOI:** 10.3390/biomedicines13040989

**Published:** 2025-04-17

**Authors:** Jun-Seok Sung, Il-Gyu Ko, Lakkyong Hwang, Sang-Hoon Kim, Jin Hee Han, Jung Won Jeon, Sae Rom Kim, Jeong Mi Lee, Cheon Woong Choi

**Affiliations:** 1Department of Physiology, College of Medicine, Kyung Hee University, Seoul 02447, Republic of Korea; jssung@hlbbiostep.com (J.-S.S.); lhwangphd@gmail.com (L.H.); 2Research Support Center, School of Medicine, Keimyung University, Deagu 42601, Republic of Korea; rhdlfrb@naver.com; 3Team of Efficacy Evaluation, Orient Genia Inc., Seongnam 13201, Republic of Korea; 4Department of Neurosurgery, Robert Wood Johnson Medical School, Rutgers, The Stat University of New Jersey, Piscataway, NJ 08901, USA; sanghoon.kim1@rutgers.edu; 5Department of Anesthesiology and Pain Medicine, Kyung Hee Medical Center, College of Medicine, Kyung Hee University, Seoul 02447, Republic of Korea; esthesi@khu.ac.kr; 6Department of Internal Medicine, Kyung Hee University Hospital at Gangdong, College of Medicine, Kyung Hee University, Seoul 05278, Republic of Korea; drglory@naver.com; 7Department of Pulmonary, Allergy and Critical Care Medicine, Kyung Hee University Hospital at Gangdong, College of Medicine, Kyung Hee University, Seoul 05278, Republic of Korea; kimsr0218@naver.com (S.R.K.); wjdal0916@naver.com (J.M.L.)

**Keywords:** particulate matter, pirfenidone, pulmonary fibrosis, inflammation, bronchoalveolar lavage fluid

## Abstract

Increased exposure to particulate matter (PM) from air pollution causes lung inflammation and increases morbidity and mortality due to respiratory diseases. Pirfenidone is an anti-fibrotic agent used to treat idiopathic pulmonary fibrosis. **Background/Objectives**: In this experiment, we studied the therapeutic effects of pirfenidone on PM-induced pulmonary fibrosis. **Methods**: Pulmonary fibrosis was induced by the intratracheal application of 100 μg/kg PM10 mixed with 200 μL saline. After 42 days of PM10 infusion, 0.2 mL of distilled water with pirfenidone was orally administered to the pirfenidone-treated groups (200 and 400 mg/kg) every other day for a total of 15 times over 30 days. **Results**: The intratracheal administration of PM resulted in lung injury and a significant decrease in the number of bronchoalveolar lavage fluid cells. PM administration increased the lung injury score, level of lung fibrosis, and production of pro-inflammatory cytokines. Pirfenidone treatment effectively suppressed transforming growth factor-β-activated kinase 1 in PM-induced pulmonary fibrosis. The present changes inhibited the expressions of mitogen-activated protein kinase kinase 3 and p38, which suppressed transforming growth factor-β, ultimately alleviating lung fibrosis. PM exposure upregulated the expressions of fibronectin and type 1 collagen. PM exposure enhanced connective tissue growth factor and hydroxyproline levels in the lung tissue. The levels of these fibrosis-related factors were inhibited by pirfenidone treatment. **Conclusions**: These results suggest that pirfenidone is therapeutically effective against PM-induced pulmonary fibrosis.

## 1. Introduction

Air pollutants consist of gaseous pollutants and particulate matter (PM). The pathophysiology of PM is determined by its origin, size, composition, solubility, and free-radical-generating ability. PM with a diameter of 10 μm (PM10) or less is known to have a more significant impact on human health, and PM10 has been shown to be associated with anencephaly in pregnant women [1]. PM concentrations continued to increase from 2010 to 2019, exceeding 10 μg/m^3^ on a global scale and surpassing 35 μg/m^3^ in more than half of the world [2,3]. Exposure to PM aggravates lung inflammation, induces oxidative stress, and causes direct toxic damage [4], which, in turn, enhances the morbidity and mortality of respiratory diseases such as asthma, pulmonary fibrosis, and chronic obstructive pulmonary disease [5]. As small PM initiates or worsens the symptoms of numerous lung diseases, it presents a growing public health threat [3].

Pulmonary fibrosis occurs when excessive collagen fibers accumulate in the lung mesenchyme, destroying the respiratory structure and reducing respiratory capacity [6]. The etiology of pulmonary fibrosis is complex, and its treatment is controversial. However, immune cells and inflammatory cytokines have been suggested as factors involved in the development of pulmonary fibrosis.

Transforming growth factor (TGF)-β is one of the cytokines that regulates cell differentiation, apoptosis, proliferation, and wound healing, and TGF-β is also a strong initiator of extracellular matrix synthesis [7]. TGF-β is known as a key mediator of pulmonary fibrosis, and the dysregulation of TGF-β expression and its action on PM-induced tissue damage causes pulmonary fibrosis progression [8]. Transforming growth factor-β-activated kinase 1 (TAK1) mediates TGF-β-induced signaling and regulates the downstream activation of mitogen-activated protein kinases (MAPKs) to control various cellular functions [9]. TAK1 is activated by environmental stress, lipopolysaccharide, tumor necrosis factor (TNF)-α, and interleukin (IL)-1 [7,10]. When the TGF-β response becomes excessive, TAK1 activity is inhibited, and there are reports that TAK1 signaling is involved in extracellular matrix processing and the progression of pulmonary fibrosis [11,12].

Despite advances in understanding pulmonary fibrosis, current treatments like corticosteroids and immune suppressants offer limited efficacy and often cause side effects. As they primarily address inflammation rather than fibrosis itself, there remains a pressing need for safer, more targeted anti-fibrotic therapies.

The active ingredient of pirfenidone, 5-methyl-1-phenyl-2-(1H)-pyridone, is an anti-fibrotic drug that inhibits the progression of fibrosis in patients with idiopathic pulmonary fibrosis. Pirfenidone is known to restore TGF-β1-activated fibroblast-induced collagen gel contractions in lung fibroblasts [13]. However, its potential to mitigate PM-induced pulmonary fibrosis remains poorly understood. Clarifying this could expand pirfenidone’s therapeutic utility and provide novel insights into air-pollution-related respiratory diseases.

In the present experiment, bronchoalveolar lavage fluid (BALF) examination was performed and the level of lung fibrosis was measured using picrosirius red staining. Lung injury scores were calculated using hematoxylin and eosin (H&E) staining. Enzyme-linked immunosorbent assay (ELISA) was used to detect the concentration levels of TNF-α, IL-1β, IL-6, TGF-β, hydroxyproline, and connective tissue growth factor (CTGF). Western blotting was conducted to calculate the protein levels of TAK1, mitogen-activated protein kinase kinase (MKK) 3, and p38. Immunohistochemical staining for collagen type I and fibronectin was performed.

## 2. Materials and Methods

### 2.1. Animals and Groups

Forty male Sprague Dawley rats weighing 200 ± 5 g (8 weeks old) were bought from a commercial breeder (Orient Co., Seongnam, Republic of Korea). The rats were divided into the following groups (n = 10 per group): control, PM10-injected, PM10-injected and 200 mg/kg pirfenidone-treated, and PM10-injected and 400 mg/kg pirfenidone-treated.

### 2.2. PM10 Injection and Pirfenidone Treatment

In the experimental animals, pulmonary fibrosis was induced by PM exposure according to the following method [14,15,16]. Anesthesia was performed using Zoletil 50^®^ (10 mg/kg, i.p.; Vibac Laboratories, Carros, France) and then 100 μg/kg PM10 in 200 μL saline was intratracheally instilled. In the control group, the same volume of saline was intratracheally instilled instead of PM10. Forty-two days after PM10 instillation, the pirfenidone-treated rats were orally administered 0.2 mL of distilled water with pirfenidone (Kolon Pharma Co., Gwacheon, Republic of Korea) every other day for a total of 15 times for 30 days (Figure 1A). Rats in the drug-naïve group were administered 0.5 mL of distilled water without pirfenidone.

### 2.3. BALF Collection and Lung Tissue Preparation

At 72 days after the PM10 injection, the rats were sacrificed. An appropriate small-bore tube was inserted into the trachea to isolate and fix it after injecting of 10 mg/kg Zoletil 50^®^ (Vibac Laboratories) into the abdominal cavity. Phosphate-buffered saline (pH 7.2) was slowly instilled into the lungs, and BALF was collected through the inserted tube. Most of the fluid was recovered (recovery rate > 90%). The right lung lobe was removed after BALF collection. Lung tissue was treated with 4% paraformaldehyde, dehydrated with graded ethanol, processed with xylene, infiltrated with paraffin, and embedded. Coronal sections of 5 μm thickness were prepared by a paraffin microtome (Thermo Co., Cheshire, UK) and then put on coated slides. Six sections were obtained from each lung tissue sample. The sides were air-dried overnight at 37 °C on hotplates.

### 2.4. BALF Cell Counting

BALF cell counts were performed using the following method [17]. The cell suspension from the BALF was mixed and diluted 1:20 with trypan blue before use. After pouring the liquid onto both sides of the hemocytometer chamber to stabilize the cells, viable cells were counted in the four corner squares.

### 2.5. Concentration of Pro-Inflammatory Cytokines and Fibrotic Factors

The levels of pro-inflammatory cytokines (TNF-α, IL-1β, and IL-6) and fibrosis-associated markers (TGF-β, hydroxyproline, and CTGF) in lung tissue were measured using ELISA. The procedure followed the manufacturer’s instructions provided with the enzyme immunoassay kit (Abcam, Cambridge, UK).

### 2.6. Hematoxylin and Eosin Staining for Lung Histopathological Evaluation

Hematoxylin and eosin (H&E) staining was performed as previously described [17] to assess lung tissue morphology. The stained lung sections were examined under a light microscope (Olympus, Tokyo, Japan) using Image-Pro^®^ Plus software (Ver. 4.5., Media Cybernetics Inc., Silver Spring, MD, USA). Lung injury was quantified following established methods [17], focusing on five distinct histopathological features: alveolar capillary congestion, hemorrhage, inflammatory cell infiltration or aggregation in the airspace, alveolar wall thickening, and hyaline membrane formation. Each parameter was independently graded on a scale from 0 to 3 (0 = absent, 1 = mild, 2 = moderate, 3 = severe).

### 2.7. Picrosirius Red Staining for Pulmonary Fibrosis Analysis

Picrosirius red staining was carried out using the following method [18]. Staining on a section was performed using a picrosirius red staining kit (Abcam) and coverslips were mounted. Using an Image-Pro^®^ plus computer-aided image analysis system (Media Cyberbetics Inc., Rockville, MD, USA) attached to a light microscope (Olympus), images of picrosirius red-stained slides were captured. Randomly, five fields were taken from each sample, expressed as images, and analyzed according to the following method [19,20]. The ratio of spots occupying parenchymal (alveolar) tissue to spots occupying collagen-positive areas was calculated. The percentage of collagen in the parenchyma of the lung sections was calculated by dividing the number of spots occupying collagen areas by the total number of spots occupying parenchymal areas.

### 2.8. Western Blotting

Western blotting was carried out using the following method [17]. Lung tissues were homogenized in lysis buffer and centrifuged at 14,000 rpm for 30 min. Next, 30 μg of protein was separated and transferred to a nitrocellular membrane. The primary and secondary antibodies used in Western blot analysis of this experiment are listed in Table 1. Most of the experimental procedures were conducted at room temperature, with the exception of the membrane transfer process, which was carried out at 4 °C. The enhanced chemiluminescence detection kit (Santa Cruz Biotechnology, Santa Cruz, CA, USA) was used for band detection. The analysis of the detected bands was performed by measuring relative protein expressions using Molecular Analyst^TM^ version 1.4.1 (Bio-Rad, Hercules, CA, USA).

### 2.9. Immunofluorescence Staining for Collagen Type I and Fibronectin

Immunofluorescence staining was carried out using the following method [21]. Paraffin slides with lung tissue were treated with xylene and graded ethanol for 5 min and washed with deionized water for 5 min. The slides were boiled in 10 mM of sodium citrate buffer at 95 °C for 2 min, stored at room temperature for 30 min, and blocked by treatment in phosphate-buffered saline with 5% normal goat serum and 0.3% Triton X-100. The tissue sections were reacted with a 1:500 dilution of mouse anti-fibronectin antibody or mouse anti-collagen type I antibody (Santa Cruz Biotechnology) overnight at 4 °C. After washing the primary antibody, the slides were incubated with secondary Alexa 488-anti-mouse IgG antibody for 2 h at room temperature. Slides were coverslip-mounted using Vectashield^®^ with 4′,6-diamidino-2-phenylindole (Vector Laboratories, Burlingame, CA, USA). Images were taken using a Leica DMi8 fluorescence microscope (Leica, Nussloch, Germany).

### 2.10. Statistical Analysis

Statistical processing of experimental data among groups was performed using one-way ANOVA and Duncan’s post hoc test with SPSS software (version 23.0, IBM Co., Armonk, NY, USA). The significance level was set at *p* < 0.05.

## 3. Results

### 3.1. BALF Cells, Lung Injury Score, and Pulmonary Fibrosis

The BALF cell counts are shown in Figure 1B. The BALF cell count was decreased by PM10-induced lung fibrosis (*p* < 0.05) but pirfenidone administration enhanced the BALF cell count (*p* < 0.05).

The evaluation of lung histopathology and the extent of pulmonary fibrosis are presented in Figure 1C. After 72 days of PM10 exposure, notable pathological changes such as intra-alveolar hemorrhage, interstitial edema, and the infiltration of inflammatory cells were evident within the alveolar spaces. Moreover, fibrotic bands or nodular structures developed in the lung tissue of the PM10-injected group. These findings demonstrate that PM10 exposure significantly aggravated both lung injury and fibrosis. However, treatment with pirfenidone alleviated the formation of fibrotic bands and nodules, leading to a marked reduction in pulmonary fibrosis (*p* < 0.05). Furthermore, pirfenidone effectively mitigated inflammatory infiltration, interstitial edema, and fibrotic structures, resulting in a significant improvement in lung injury scores (*p* < 0.05).

### 3.2. Pro-Inflammatory Cytokines

The levels of TNF-α, IL-1β, and IL-6 expression, which are called pro-inflammatory cytokines, in lung tissue are presented in Figure 2. The levels of TNF-α, IL-1β, and IL-6 expression were enhanced in PM10-induced pulmonary fibrosis (*p* < 0.05). However, pirfenidone application inhibited the levels of TNF-α, IL-1β, and IL-6 expression (*p* < 0.05). These findings suggest that pirfenidone effectively suppresses the inflammatory cascade triggered by PM10 and may play a protective role in mitigating inflammation-associated pulmonary fibrosis.

### 3.3. Fibrotic Factors

The levels of TGF-β, hydroxyproline, and CTGF expression, which are named as fibrotic factors, in lung tissue are presented in Figure 3. PM10-injection-induced lung fibrosis enhanced the levels of TGF-β, hydroxyproline, and CTGF expression (*p* < 0.05), but pirfenidone administration decreased the levels of TGF-β, hydroxyproline, and CTGF expression (*p* < 0.05). These results indicate that pirfenidone effectively suppresses fibrogenic signaling and collagen synthesis, suggesting its therapeutic potential in attenuating PM10-induced lung fibrosis.

### 3.4. TAK1/MKK3/p38 Signaling Pathway

The expression levels of TAK1/MKK3/p38 are presented in Figure 4. PM10-injection-induced lung fibrosis enhanced TAK1/MKK3/p38 expression (*p* < 0.05). Pirfenidone administration significantly decreased their expression levels (*p* < 0.05), suggesting its role in suppressing this pro-fibrotic signaling pathway.

### 3.5. Collagen Type I

The collagen type I expression detected by immunohistochemistry in the lung tissue is presented in Figure 5. PM10-induced pulmonary fibrosis enhanced collagen type I expression in lung tissue, but pirfenidone administration suppressed collagen type I expression, suggesting its anti-fibrotic effect on collagen deposition.

### 3.6. Fibronectin

The fibronectin expression detected by immunohistochemistry in the lung tissue is presented in Figure 6. PM10-induced pulmonary fibrosis increased fibronectin expression in lung tissue, but pirfenidone administration suppressed fibronectin expression, indicating its inhibitory effect on fibrotic remodeling.

## 4. Discussion

Small PM induced inflammation and oxidative stress in mouse macrophages [22]. Macrophages are a sensitive and powerful defense mechanism against harmful factors, including PMs. Long-term accumulation and exposure to PM lead to macrophage activation, which secretes pro-inflammatory cytokines and produces cytotoxic reactive oxygen species, proteases, and bioactive lipids [23]. The lung injury model used in this study is known to induce inflammatory mediator expression, neutrophil accumulation, and alveolar damage after PM deposition. In addition, the long-term accumulation of PM in the lungs eventually leads to pulmonary fibrosis [14,15,16]. In comparison with human lung pathologies associated with PM exposure, the lung injuries observed in our mouse model with PM10 instillation closely resemble early-stage acute lung injury, characterized by alveolar inflammation and the disruption of the epithelial barrier. When combined with fibrogenic factors, PM10 exposure may also induce fibrotic changes in the lung parenchyma, partially reflecting early features of pulmonary fibrosis in humans [24,25,26]. These findings establish PM10 as a relevant agent for modeling environmentally triggered fibrotic lung disease and for evaluating therapeutic interventions such as pirfenidone. PM10 was used in this study to model pulmonary fibrosis because it can penetrate deep into the lungs and trigger oxidative stress, inflammation, and fibrotic responses. In the present study, the intratracheal administration of PM induced alveolar capillary congestion, the infiltration of inflammatory cells, and pulmonary fibrosis. These pathological changes led to a higher lung injury score compared to the non-PM-treated group.

Likewise, the present results demonstrated that the lung damage caused by PM administration may be due to an increase in TNF-α, IL-1β, and IL-6—the pro-inflammatory cytokines—in lung tissue. An increase in the BALF cell count was also a consequence of these changes. That is, when PM accumulates in the lungs, pro-inflammatory cytokines are overproduced, which leads to an excessive increment in BALF cell numbers.

Pirfenidone, an anti-fibrotic agent, has been shown to suppress the production of pro-inflammatory cytokines and chemokines, thereby reducing inflammatory responses and T-cell activity [27]. Additionally, pirfenidone inhibits the expression of intracellular adhesion molecule-1 and promotes the expression of anti-inflammatory cytokines [28,29]. In the current study, pirfenidone significantly reduced the production of pro-inflammatory cytokines in the lungs, further confirming its anti-inflammatory properties.

PM exposure leads to marked inflammation, which over time results in pulmonary fibrosis and lung damage [30,31]. The previous study demonstrated that fibrosis was progressively aggravated in mice exposed to PM for 8 weeks, suggesting that PM exposure directly promotes fibrosis [31]. However, the precise mechanisms underlying PM-induced lung fibrosis remain unclear. One potential pathway is the TAK1/MKK3/p38 signaling cascade, which regulates TGF-β, a major mediator of fibrosis [32,33]. TAK1 has been shown to play a critical role in the activation of TGF-β-induced MKK3 and subsequent collagen production in mesodermal cells [29]. Additionally, the expression of TGF-β-induced fibronectin in fibroblasts is mediated by TAK1, and TAK1-deficient fibroblasts exhibit a reduced pro-fibrotic response to TGF-β stimulation [7,32,33]. Notably, the pro-fibrotic function of TAK1 signaling has been observed in lungs exposed to PM, where TAK1 activation leads to the increased phosphorylation of p38 MAPK, promoting fibrotic events in lung tissue [32,33].

In the current study, pirfenidone administration effectively regulated TAK1 in PM-induced pulmonary fibrosis. This is consistent with previous findings that pirfenidone treatment has an inhibitory effect on TAK1 expression [34]. Furthermore, these changes lead to the suppression of TGF-β by regulating MKK3 and p38 expression, which suggests the alleviation of fibrotic conditions. Collagen and fibronectin, which are extracellular matrix proteins, are major components of fibrotic conditions and also play a key role in the progression of fibrotic lesions [35]. TAK1 is involved in the expression of type 1 collagen and fibronectin induced by TGF-β activating signaling cascades [7,36].

In the present results, we evaluated the expression levels of fibrotic factors, including TGF-β, hydroxyproline, and CTGF, in the lung tissue of rats with PM-induced pulmonary fibrosis. The TGF-β, hydroxyproline, and CTGF levels in lung tissues were significantly increased by PM exposure. The expression levels of fibronectin and type 1 collagen were also significantly upregulated by PM exposure, but pirfenidone treatment suppressed the expression of these factors. The current experiment is similar to previous studies in that we found that pirfenidone treatment modulates the effects of fibrotic factors such as TGF-β, hydroxyproline, CTGF, fibronectin, and type 1 collagen in lung tissue [37,38,39]. A reduction in these fibrotic factors ultimately results in a decrease in the extent of fibrotic areas in lung tissue. These results suggest that pirfenidone is therapeutically effective against PM-induced pulmonary fibrosis. However, lung function tests including arterial blood gas analysis after PM exposure were not conducted in this study. For this reason, further studies using various lung function tests are needed to improve our understanding of PM-induced fibrosis.

## 5. Conclusions

This study demonstrates that PM exposure induces significant lung damage, characterized by inflammation, oxidative stress, and pulmonary fibrosis, primarily through the activation of pro-inflammatory cytokines and the TAK1/MKK3/p38 signaling pathway. Pirfenidone effectively attenuated these pathological changes by suppressing pro-inflammatory cytokines, fibrotic factors, and downregulating TAK1 signaling. These findings suggest that pirfenidone holds therapeutic potential in mitigating PM-induced pulmonary fibrosis and may serve as a promising intervention for preventing PM-related lung injury.

## Figures and Tables

**Figure 1 biomedicines-13-00989-f001:**
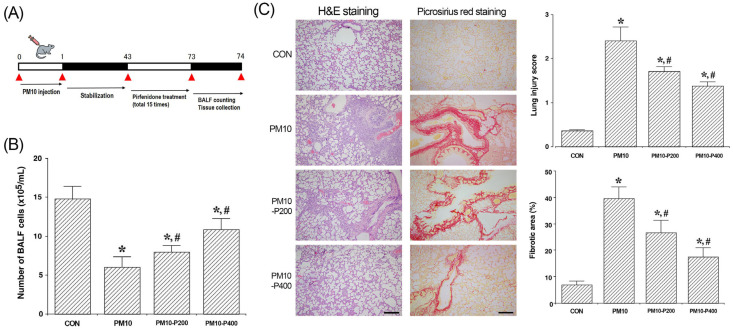
Changes in bronchoalveolar lavage fluid (BALF) cells, lung injury score, and pulmonary fibrotic level. (**A**) Experimental schedule. (**B**) Number of BALF cells. (**C**) Lung histopathological evaluation with hematoxylin and eosin (H&E) staining and pulmonary fibrotic area with picrosirius red staining. Red color is fibrotic response. Scale bar is 150 μm. CON, control group; PM10, PM10-injected group; PM10-P200, PM10-injected and 200 mg/kg pirfenidone-treated group; PM10-P400, PM10-injected and 400 mg/kg pirfenidone-treated group. * shows *p* < 0.05 when compared to control group. # shows *p* < 0.05 when compared to PM10-injected group.

**Figure 2 biomedicines-13-00989-f002:**
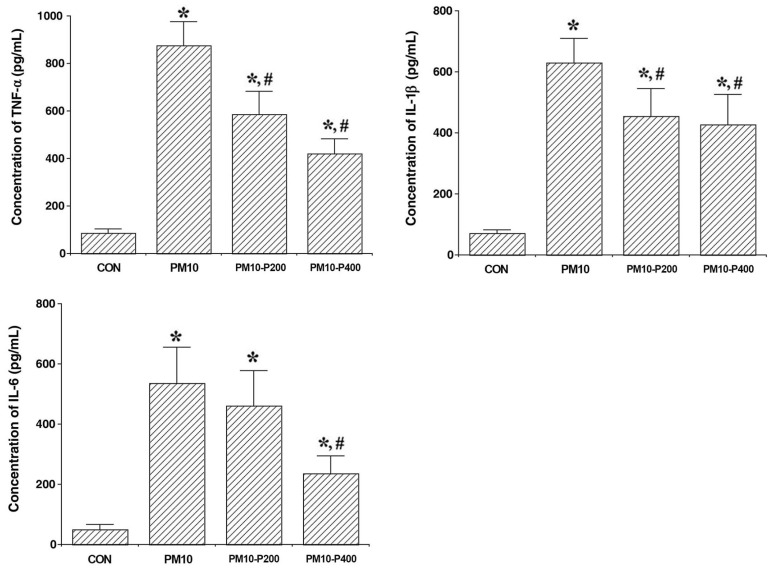
Changes in pro-inflammatory cytokine concentrations. **Upper-left panel**: Concentration of tumor necrosis factor (TNF)-α. **Upper-right panel**: Concentration of interleukin (IL)-1β. **Lower-left**: Concentration of IL-6. CON, control group; PM10, PM10-injected group; PM10-P200, PM10-injected and 200 mg/kg pirfenidone-treated group; PM10-P400, PM10-injected and 400 mg/kg pirfenidone-treated group. * shows *p* < 0.05 when compared to control group. # shows *p* < 0.05 when compared to PM10-injected group.

**Figure 3 biomedicines-13-00989-f003:**
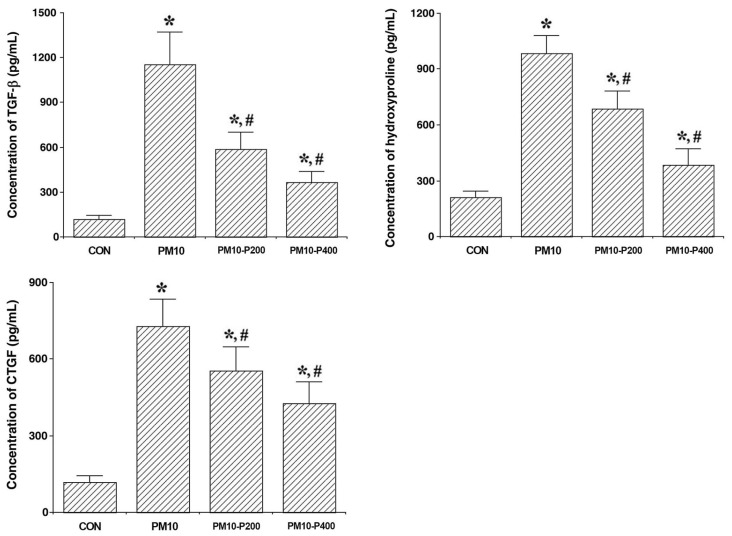
Changes in fibrotic factors concentrations. **Upper-left panel**: Concentration of transforming growth factor (TGF)-β. **Upper-right panel**: Concentration of hydroxyproline **Lower-left**: Level of connective tissue growth factor (CTGF). CON, control group; PM10, PM10-injected group; PM10-P200, PM10-injected and 200 mg/kg pirfenidone-treated group; PM10-P400, PM10-injected and 400 mg/kg pirfenidone-treated group. * shows *p* < 0.05 when compared to control group. # shows *p* < 0.05 when compared to PM10-injected group.

**Figure 4 biomedicines-13-00989-f004:**
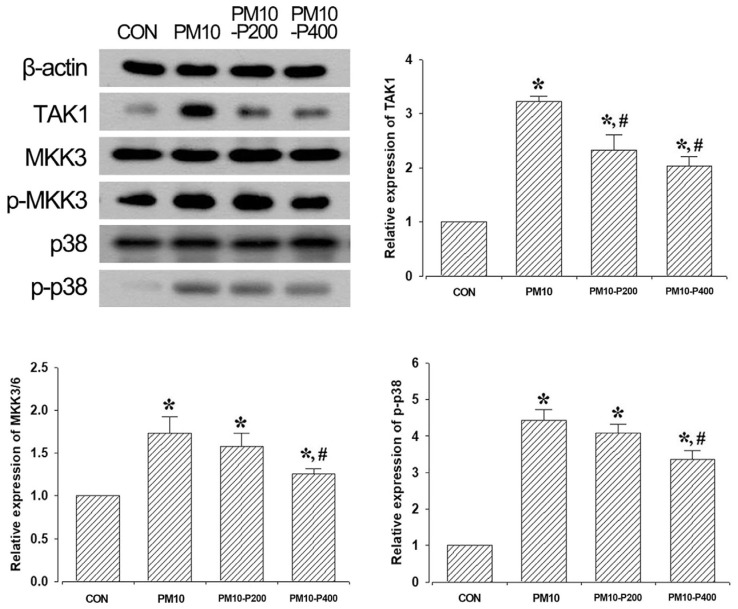
Expressions of transforming growth factor-β-activated kinase 1 (TAK1), mitogen-activated protein kinase kinase 3 (MKK3), and p38. **Upper-left panel**: Representative expression of TAK1, MKK3, and p38. **Upper-right panel**: Relative expression of TAK1. **Lower-left panel**: Relative expression of phosphorylated MKK3. **Lower-left panel**: Relative expression of phosphorylated p38. CON, control group; PM10, PM10-injected group; PM10-P200, PM10-injected and 200 mg/kg pirfenidone-treated group; PM10-P400, PM10-injected and 400 mg/kg pirfenidone-treated group. * shows *p* < 0.05 when compared to control group. # shows *p* < 0.05 when compared to PM10-injected group.

**Figure 5 biomedicines-13-00989-f005:**
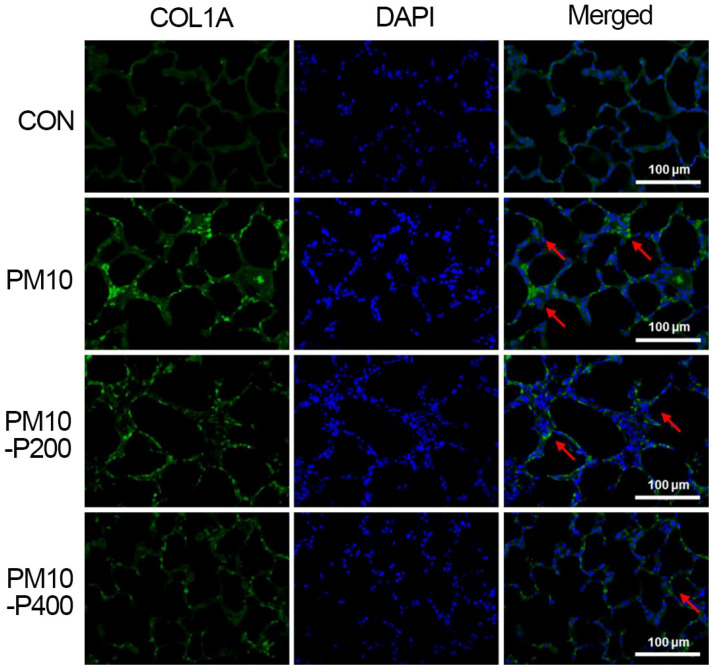
Collagen type I (COL1A) expression in lung tissue. Collagen type I expression (green color). Counting staining with 4′,6-diamidino-2-phenylindol (DAPI, blue color). Red arrow represents collagen type I-positive cells. CON, control group; PM10, PM10-injected group; PM10-P200, PM10-injected and 200 mg/kg pirfenidone-treated group; PM10-P400, PM10-injected and 400 mg/kg pirfenidone-treated group.

**Figure 6 biomedicines-13-00989-f006:**
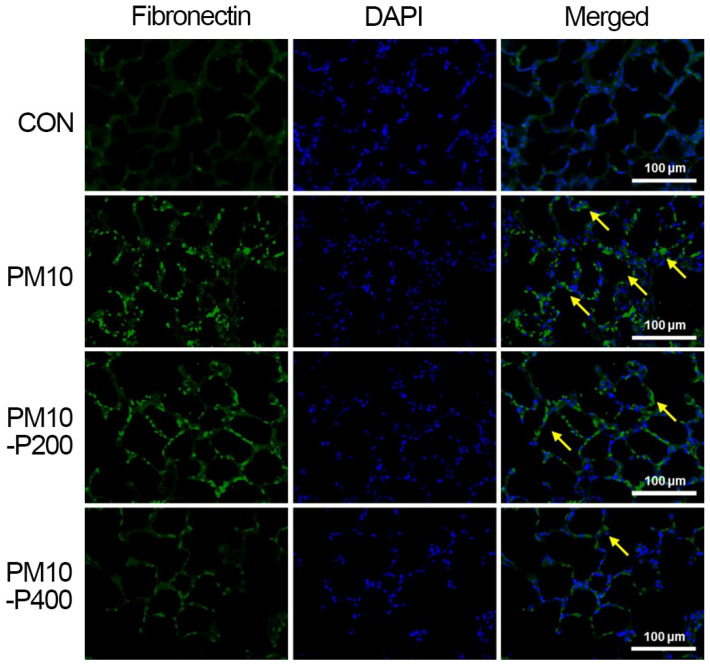
Fibronectin expression in lung tissue. Fibronectin expression (green color). Counting staining with 4′,6-diamidino-2-phenylindol (DAPI, blue color). Yellow arrow represents fibronectin-positive cells. CON, control group; PM10, PM10-injected group; PM10-P200, PM10-injected and 200 mg/kg pirfenidone-treated group; PM10-P400, PM10-injected and 400 mg/kg pirfenidone-treated group.

**Table 1 biomedicines-13-00989-t001:** Primary and secondary antibodies used for Western blot analysis.

Classification	Items	Source	Titer	Company
Primary antibody	TAK1, p-MKK3, MKK3, p38, p-p38	Anti-rabbit	1:1000	Cell Signaling Technology, Danvers, MA, USA
	β-actin	Anti-mouse	1:1000	Santa Cruz Biotechnology, Santa Cruz, CA, USA
Secondary antibody	HRP-conjugated IgG	Mouse	1:2000	Vector Laboratories, Burlingame, CA, USA
		Rabbit		

TAK1, transforming growth factor-β-activated kinase 1; MKK3, mitogen-activated protein kinase kinase 3; p, phosphorylated; HRP, horseradish peroxidase.

## Data Availability

The data are presented in the manuscript. Additional information obtained during the experiments is available upon request.

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
