# Peer review of "Pirfenidone Alleviates Against Fine Particulate Matter-Induced Pulmonary Fibrosis Modulating via TGF-β1/TAK1/MKK3/p38 MAPK Signaling Pathway in Rats"

_biomedicines, 2025, doi:10.3390/biomedicines13040989_

Round 1
Reviewer 1 Report
Comments and Suggestions for Authors
The Authors describe in the article entitled "Pirfenidone alleviates against fine particulate matter-induced pulmonary fibrosis modulating via TGF-β1/TAK1/MKK3/ p38MAPK signaling pathway in rats" describe the damage caused by PM10 at the pulmonary level causing fibrosis through a series of processes involving macrophages, cytokines, IL1 and IL6. Interesting in this study is the analysis of the BALF where a good part of these processes that cause pulmonary fibrosis are highlighted.
The study is a basic research applied at the level of 40 rats and highlights a clear text and figures that make reading the article even easier.
The conclusions also concern the efficacy of pirfenidone capable of reducing pulmonary fibrosis as an antifibrotic in rats exposed to PM10
Author Response
Answers to Reviewers’ Comments
Journal name: Biomedicines
Manuscript ID: biomedicines-3516725
Type of manuscript: Article
Title: Pirfenidone alleviates against fine particulate matter-induced pulmonary fibrosis modulating via TGF-β1/TAK1/MKK3/ p38MAPK signaling pathway in rats
Authors: Jun-Seok Sung, Il-Gyu Ko, Lakkyong Hwang, Sang Hoon Kim, Jin Hee Han, Jung Won Jeon, Sae Rom Kim, Jeong Mi Lee, Cheon Woong Choi *
We sincerely appreciate for your kind advice and comment to our manuscript. We revised to manuscript according to the reviewer’s comments. We added new experimental data, and modifications were expressed in red.
Reviewer -1
The authors describe in the article entitled "Pirfenidone alleviates against fine particulate matter-induced pulmonary fibrosis modulating via TGF-β1/TAK1/MKK3/p38MAPK signaling pathway in rats" describe the damage caused by PM10 at the pulmonary level causing fibrosis through a series of processes involving macrophages, cytokines, IL1 and IL6.
Major comments
Q1. Interesting in this study is the analysis of the BALF where a good part of these processes that causes pulmonary fibrosis are highlighted. The study is a basic research applied at the level of 40 rats and highlights a clear text and figures that make reading the article even easier. The conclusions also concern the efficacy of pirfenidone capable of reducing pulmonary fibrosis as an anti-fibrotic in rats exposed to PM10.
Author answer: First all, thank you for your kind advice and comments. This study aimed to evaluate the potential of pirfenidone as a treatment for pulmonary fibrosis induced by exposure to fine dust (PM10). In response to the reviewer’s comments, we conducted additional analysis on changes in BALF as well as on inflammatory and fibrotic markers using an experimental rat model. Our findings support the anti-fibrotic efficacy of pirfenidone, demonstrating its ability to reduce pulmonary fibrosis in rats exposed to PM10 and highlighting its potential as a therapeutic agent. Air pollution, including fine dust, continues to pose a significant global health challenge. We hope our research contributes meaningfully to the understanding and treatment of diseases caused by such environmental factors.
Reviewer 2 Report
Comments and Suggestions for Authors
The topic of the article is very interesting
1-This model of injury is similar to which lung injury in the human? it is added in the discussion part
2-Lung pathology study of experimental groups should be reported .
3-What is the lung function test index in your study?
4-The discussion section is written very briefly. please explain
completely
5-The conclusion of the study is not written.
6- what is limitation of your study.
Comments on the Quality of English LanguageThe English could be improved to more clearly express the research
Author Response
Answers to Reviewers’ Comments
Journal name: Biomedicines
Manuscript ID: biomedicines-3516725
Type of manuscript: Article
Title: Pirfenidone alleviates against fine particulate matter-induced pulmonary fibrosis modulating via TGF-β1/TAK1/MKK3/ p38MAPK signaling pathway in rats
Authors: Jun-Seok Sung, Il-Gyu Ko, Lakkyong Hwang, Sang Hoon Kim, Jin Hee Han, Jung Won Jeon, Sae Rom Kim, Jeong Mi Lee, Cheon Woong Choi *
We sincerely appreciate for your kind advice and comment to our manuscript. We revised to manuscript according to the reviewer’s comments. We added new experimental data, and modifications were expressed in red.
Reviewer -2
The topic of the article is very interesting
Major comments
Q1. This model of injury is similar to which lung injury in the human? it is added in the discussion part
Author answer: First all, thank you for your kind advice and comments. According to reviewer’s comment, we added new sentence and reference was inserted in manuscript.
- Following sentence were added to the Discussion part
In comparison with human lung pathologies associated with PM exposure, lung injury observed in a mouse model with PM10 instillation closely resembles early-stage acute lung injury (ALI), characterized by alveolar inflammation and disruption of the epithelial barrier. When combined with fibrogenic factors, PM10 exposure may also induce fibrotic changes in the lung parenchyma, partially reflecting early features of pulmonary fibrosis in humans [37-39].
- Following reference were added to the Reference part
- Valderrama, A.; Ortiz-Hernández, P.; Agraz-Cibrián, J.M.; Tabares-Guevara, J.H.; Gómez, D.M.; Zambrano-Zaragoza, J.F.; Taborda, N.A.; Hernandez, J.C. Particulate matter (PM10) induces in vitro activation of human neutrophils, and lung histopathological alterations in a mouse model. Sci. Rep. 2022, 9, 7581. http://doi.org/10.1038/s41598-022-11553-6.
- Park, S.Y.; An, K.S.; Lee, B.; Kang, J.H.; Jung, H.J.; Kim, M.W.; Ryu, H.Y.; Shim, K.S.; Nam, K.T.; Yoon, Y.S.; Oh, S.H. Establishment of particulate matter-induced lung injury model in mouse. Lab. Anim. Res. 2021, 30, 20. http://doi.org/10.1186/s42826-021-00097-x.
- Han, H.; Oh, E.Y.; Lee, J.H.; Park, J.W.; Park, H.J. Effects of Particulate Matter 10 Inhalation on Lung Tissue RNA expression in a Murine Model. Tuberc. Respir. Dis (Seoul). 2021, 84, 55-66. http://doi.org/10.4046/trd.2020.0107.
Q2. Lung pathology study of experimental groups should be reported.
Author answer: According to reviewer’s comment, we additionally performed H&E staining and histopathological analysis. Based on new analysis results, the modified figure, sentence, and adding references were inserted in manuscript.
- Following sentence were added to the Material & methods part
2.5. Hematoxylin & eosin staining for lung histopathological evaluation
Hematoxylin and eosin (H&E) staining was performed as previously described [17] to assess lung tissue morphology. Stained lung sections were examined under a light microscope (Olympus, Tokyo, Japan) using Image-Pro® Plus software (Media Cybernetics Inc., Silver Spring, MD, USA). Lung injury was quantified following established methods [17], focusing on five distinct histopathological features: alveolar capillary congestion, hemorrhage, inflammatory cell infiltration or aggregation in the airspace, alveolar wall thickening, and hyaline membrane formation. Each parameter was independently graded on a scale from 0 to 3 (0 = absent, 1 = mild, 2 = moderate, 3 = severe).
- Following sentence were added to the Results part
3.1. BALF cells, lung injury score, and pulmonary fibrosis
The BALF cell counts are shown in Figure 1B. The BALF cell count was decreased by PM10-induced lung fibrosis (P<0.05). But, pirfenidone administration enhanced the BALF cell count (P<0.05).
The evaluation of lung histopathology and the extent of pulmonary fibrosis is presented in Figure 1C. After 72 days of PM10 exposure, notable pathological changes such as intra-alveolar hemorrhage, interstitial edema, and infiltration of inflammatory cells were evident within the alveolar spaces. Moreover, fibrotic bands or nodular structures developed in the lung tissue of the PM10-injected group. These findings demonstrate that PM10 exposure significantly aggravated both lung injury and fibrosis. However, treatment with pirfenidone alleviated the formation of fibrotic bands and nodules, leading to a marked reduction in pulmonary fibrosis (P<0.05). Furthermore, pirfenidone effectively mitigated inflammatory infiltration, interstitial edema, and fibrotic structures, resulting in a significant improvement in lung injury scores (P<0.05).
- Following figure and figure legends was changed to the Results part
Figure 1. Changes of bronchoalveolar lavage fluid (BALF) cells, lung injury score, and pulmonary fibrotic level. (A) Experimental schedule. (B) Number of BALF cells. (C) Lung histopathological evaluation with hemaytoxylin & eosin (H&E) staining and pulmonary fibrotic area with picrosirius red staining. Red color is fibrotic response.
- Following sentence were added to the Discussion part
In the present study, intratracheal administration of PM induced alveolar capillary congestion, infiltration of inflammatory cells, and pulmonary fibrosis. These pathological changes led to a higher lung injury score compared to the non-PM-treated group.
Likewise, the present results demonstrated that the lung damage caused by PM administration may be due to be an increase in TNF-α, IL-1β, and IL-6, the pro-inflammatory cytokines, in lung tissue.
- Following reference were added to the Reference part
- Ko, I.G.; Hwang, J.J.; Chang, B.S.; Kim, S.H.; Jin, J.J.; Hwang, L.; Kim, C.J.; Choi, C.W. Polydeoxyribonucleotide ameliorates lipopolysaccharide-induced acute lung injury via modulation of the MAPK/NF-κB signaling pathway in rats. Int. Immunopharmacol. 2020, 83, 106444. https://doi.org/10.1016/j.intimp.2020.106444
- Following sentence were deleted to the Material & methods part
By an Image-Pro® plus computer-aided image analysis system (Media Cyberbetics Inc., Silver Spring, MD, USA) attached to a light microscope (Olympus, Tokyo, Japan), images of picrosirius red-stained slides were captured.
Q3. What is the lung function test index in your study?
Author answer: In this study, we evaluated the pathological changes in BALF and lung tissue, and the changes in inflammation and fibrosis through molecular biological evaluation. Unfortunately, in this study, there were no lung function results evaluated during the survival of experimental animals. Typically, gas analysis through breath analysis or analysis evaluating arterial blood is performed, but this lung function analysis was not performed in this study. We consider this very regrettable and will mention it as a limitation of the study in the discussion section. We will also try to analyze lung function in future studies.
- Following sentence were added to the Discussion part
However, lung function test including arterial blood gas analysis after PM exposure was not measured in this study. For this reason, further studies using various lung function tests are needed to improve PM-induced fibrosis.
Q4. The discussion section is written very briefly. Please explain completely
Author answer: According to reviewer’s comment, we added new results to the discussion part and comprehensively supplemented it.
- Following sentence were added to the Discussion part
In comparison with human lung pathologies associated with PM exposure, lung injury observed in a mouse model with PM10 instillation closely resembles early-stage acute lung injury (ALI), characterized by alveolar inflammation and disruption of the epithelial barrier. When combined with fibrogenic factors, PM10 exposure may also induce fibrotic changes in the lung parenchyma, partially reflecting early features of pulmonary fibrosis in humans [24-26].
In the present study, intratracheal administration of PM induced alveolar capillary congestion, infiltration of inflammatory cells, and pulmonary fibrosis. These pathological changes led to a higher lung injury score compared to the non-PM-treated group.
Likewise, the present results demonstrated that the lung damage caused by PM administration may be due to be an increase in TNF-α, IL-1β, and IL-6, the pro-inflammatory cytokines, in lung tissue.
These results suggest that pirfenidone is therapeutically effective against PM-induced pulmonary fibrosis. However, lung function test including arterial blood gas analysis after PM exposure was not measured in this study. For this reason, further studies using various lung function tests are needed to improve PM-induced fibrosis.
Q5. The conclusion of the study is not written.
Author answer: According to reviewer comment, we created a new Conclusion part and added a summary.
- Conclusion
This study demonstrates that PM exposure induces significant lung damage characterized by inflammation, oxidative stress, and pulmonary fibrosis, primarily through the activation of pro-inflammatory cytokines and the TAK1/MKK3/p38 signaling pathway. Pirfenidone effectively attenuated these pathological changes by suppressing pro-inflammatory cytokines, fibrotic factors, and downregulating TAK1 signaling. These findings suggest that pirfenidone holds therapeutic potential in mitigating PM-induced pulmonary fibrosis and may serve as a promising intervention for preventing PM-related lung injury.
Q6. What is limitation of your study.
Author answer: According to reviewer comment, we have added a sentences related to the limitations of this study to the discussion part.
- Following sentence were added to the Discussion part
However, lung function test including arterial blood gas analysis after PM exposure was not measured in this study. For this reason, further studies using various lung function tests are needed to improve PM-induced fibrosis.
Reviewer 3 Report
Comments and Suggestions for Authors
Recommendations are as follows :
1. The use of PM10 for pulmonary fibrosis modeling method has no literature basis, if the study of unique innovation, it is recommended to elaborate in the article ;
2.2.3 and 2.5 part of the experimental operation content writing is consistent with the previous part, it is recommended to omit ;
3.It is recommended to increase pathological staining experiments such as HE and Masson staining ;
4. The result part of the article is only a simple list of results. It is recommended that the results and discussion parts be written in detail and increased in depth.
Author Response
Answers to Reviewers’ Comments
Journal name: Biomedicines
Manuscript ID: biomedicines-3516725
Type of manuscript: Article
Title: Pirfenidone alleviates against fine particulate matter-induced pulmonary fibrosis modulating via TGF-β1/TAK1/MKK3/ p38MAPK signaling pathway in rats
Authors: Jun-Seok Sung, Il-Gyu Ko, Lakkyong Hwang, Sang Hoon Kim, Jin Hee Han, Jung Won Jeon, Sae Rom Kim, Jeong Mi Lee, Cheon Woong Choi *
We sincerely appreciate for your kind advice and comment to our manuscript. We revised to manuscript according to the reviewer’s comments. We added new experimental data, and modifications were expressed in red.
Reviewer-3
Recommendations are as follows :
Q1. The use of PM10 for pulmonary fibrosis modeling method has no literature basis, if the study of unique innovation, it is recommended to elaborate in the article.
Author answer: First all, thank you for your kind advice and comments. We some agree the reviewer’s observation regarding the limited number of studies supporting the use of PM10 in pulmonary fibrosis modeling. Although PM10 has been widely investigated for its involvement in respiratory conditions such as inflammation, COPD, and asthma, its specific contribution to fibrotic lung pathology is not yet well-established in experimental models. In contrast to well-characterized inducers like bleomycin or silica, the application of PM10 in this context remains relatively underexplored. However, several recent studies have indicated that chronic exposure to ambient particulate matter, including PM10, may contribute to fibrotic changes in the lung.
In the previous study, Han et al. (2021) reported that murine models where PM10 inhalation led to airway inflammation and fibrosis. The research highlighted that PM10 exposure resulted in significant inflammatory responses and fibrotic developments in the lung tissues of mice.
Other study showed that the effect of PM10 on human neutrophils and in mice. The findings revealed that PM10 exposure induced cytotoxic effects on neutrophils and led to inflammatory cell infiltration in mouse lung tissues, suggesting a mechanism by which PM10 contributes to pulmonary fibrosis (Valderrama et al., 2022). Furthermore, in a cohort study, Winterbottom et al. (2017) has shown that ambient air pollution, measured by average PM₁₀ concentration, is associated with an increased rate of decline in forced vital capacity (FVC) in individuals with idiopathic pulmonary fibrosis. This association indicates a potential mechanistic role for PM10 in the progression of pulmonary fibrosis. These understanding, our study aims to apply PM10 administrate in a controlled mouse model to evaluate its potential as an inducer or accelerator of pulmonary fibrosis. This approach, while not yet standardized, provides novel insights into environmental pollutant-associated fibrogenesis, which may be especially relevant for populations exposed to high urban air pollution. According to reviewer comment, we added a sentences and reference to the manuscript.
[Reference]
Han, H.; Oh, E.Y.; Lee, J.H.; Park, J.W.; Park, H.J. Effects of Particulate Matter 10 Inhalation on Lung Tissue RNA expres-sion in a Murine Model. Tuberc. Respir. Dis (Seoul). 2021, 84, 55-66. http://doi.org/10.4046/trd.2020.0107.
Valderrama, A.; Ortiz-Hernández, P.; Agraz-Cibrián, J.M.; Tabares-Guevara, J.H.; Gómez, D.M.; Zambrano-Zaragoza, J.F.; Taborda, N.A.; Hernandez, J.C. Particulate matter (PM10) induces in vitro activation of human neutrophils, and lung histopathological alterations in a mouse model. Sci. Rep. 2022, 9, 7581. http://doi.org/10.1038/s41598-022-11553-6.
Winterbottom, C.J.; Shah, R.J.; Patterson, K.C.; Kreider, M.E.; Panettieri, R.A.; Rivera-Lebron, B.; Miller, W.T.; Litzky, L.A.; Penning, T.M.; Heinlen, K.; Jackson. T.; Localio. A.R.; Christie, J.D. Exposure to ambient particulate matter is associated with accelerated functional decline in idiopathic pulmonary fibrosis. Chest. 2018, 153, 1221-1228. http://doi.org/10.1016/j.chest.2017.07.034.
- Following sentence were added to the Discussion part
In comparison with human lung pathologies associated with PM exposure, lung injury observed in a mouse model with PM10 instillation closely resembles early-stage acute lung injury (ALI), characterized by alveolar inflammation and disruption of the epithelial barrier. When combined with fibrogenic factors, PM10 exposure may also induce fibrotic changes in the lung parenchyma, partially reflecting early features of pulmonary fibrosis in humans [24-26].
- Following reference were added to the Reference part
- Valderrama, A.; Ortiz-Hernández, P.; Agraz-Cibrián, J.M.; Tabares-Guevara, J.H.; Gómez, D.M.; Zambrano-Zaragoza, J.F.; Taborda, N.A.; Hernandez, J.C. Particulate matter (PM10) induces in vitro activation of human neutrophils, and lung histopathological alterations in a mouse model. Sci. Rep. 2022, 9, 7581. http://doi.org/10.1038/s41598-022-11553-6.
- Park, S.Y.; An, K.S.; Lee, B.; Kang, J.H.; Jung, H.J.; Kim, M.W.; Ryu, H.Y.; Shim, K.S.; Nam, K.T.; Yoon, Y.S.; Oh, S.H. Establishment of particulate matter-induced lung injury model in mouse. Lab. Anim. Res. 2021, 30, 20. http://doi.org/10.1186/s42826-021-00097-x.
- Han, H.; Oh, E.Y.; Lee, J.H.; Park, J.W.; Park, H.J. Effects of Particulate Matter 10 Inhalation on Lung Tissue RNA expression in a Murine Model. Tuberc. Respir. Dis (Seoul). 2021, 84, 55-66. http://doi.org/10.4046/trd.2020.0107.
Q2. 2.2.3 and 2.5 part of the experimental operation content writing is consistent with the previous part, it is recommended to omit
Author answer: Thank you for pointing out my mistake. We have removed the duplicate part according to the reviewer comments. And then, we modified the content to fit the research method.
- Following sentence were changed to the Methods part
2.5. Concentration of pro-inflammatory cytokines and fibrotic factors
The levels of pro-inflammatory cytokines (TNF-α, IL-1β, and IL-6) and fibrosis-associated markers (TGF-β, hydroxyproline, and CTGF) in lung tissue were measured using ELISA. The procedure followed the manufacturer's instructions provided with the enzyme immunoassay kit (Abcam, Cambridge, UK).
Q3. It is recommended to increase pathological staining experiments such as HE and Masson staining
Author answer: According to reviewer’s comment, we additionally performed H&E staining and histopathological analysis. Based on new analysis results, the modified figure, sentence, and adding references were inserted in manuscript.
- Following sentence were added to the Material & methods part
2.5. Hematoxylin & eosin staining for lung histopathological evaluation
Hematoxylin and eosin (H&E) staining was performed as previously described [17] to assess lung tissue morphology. Stained lung sections were examined under a light microscope (Olympus, Tokyo, Japan) using Image-Pro® Plus software (Media Cybernetics Inc., Silver Spring, MD, USA). Lung injury was quantified following established methods [17], focusing on five distinct histopathological features: alveolar capillary congestion, hemorrhage, inflammatory cell infiltration or aggregation in the airspace, alveolar wall thickening, and hyaline membrane formation. Each parameter was independently graded on a scale from 0 to 3 (0 = absent, 1 = mild, 2 = moderate, 3 = severe).
- Following sentence were added to the Results part
3.1. BALF cells, lung injury score, and pulmonary fibrosis
The BALF cell counts are shown in Figure 1B. The BALF cell count was decreased by PM10-induced lung fibrosis (P<0.05). But, pirfenidone administration enhanced the BALF cell count (P<0.05).
The evaluation of lung histopathology and the extent of pulmonary fibrosis is presented in Figure 1C. After 72 days of PM10 exposure, notable pathological changes such as intra-alveolar hemorrhage, interstitial edema, and infiltration of inflammatory cells were evident within the alveolar spaces. Moreover, fibrotic bands or nodular structures developed in the lung tissue of the PM10-injected group. These findings demonstrate that PM10 exposure significantly aggravated both lung injury and fibrosis. However, treatment with pirfenidone alleviated the formation of fibrotic bands and nodules, leading to a marked reduction in pulmonary fibrosis (P<0.05). Furthermore, pirfenidone effectively mitigated inflammatory infiltration, interstitial edema, and fibrotic structures, resulting in a significant improvement in lung injury scores (P<0.05).
- Following figure and figure legends was changed to the Results part
Figure 1. Changes of bronchoalveolar lavage fluid (BALF) cells, lung injury score, and pulmonary fibrotic level. (A) Experimental schedule. (B) Number of BALF cells. (C) Lung histopathological evaluation with hemaytoxylin & eosin (H&E) staining and pulmonary fibrotic area with picrosirius red staining. Red color is fibrotic response.
- Following sentence were added to the Discussion part
In the present study, intratracheal administration of PM induced alveolar capillary congestion, infiltration of inflammatory cells, and pulmonary fibrosis. These pathological changes led to a higher lung injury score compared to the non-PM-treated group.
Likewise, the present results demonstrated that the lung damage caused by PM administration may be due to be an increase in TNF-α, IL-1β, and IL-6, the pro-inflammatory cytokines, in lung tissue.
- Following reference were added to the Reference part
- Ko, I.G.; Hwang, J.J.; Chang, B.S.; Kim, S.H.; Jin, J.J.; Hwang, L.; Kim, C.J.; Choi, C.W. Polydeoxyribonucleotide ameliorates lipopolysaccharide-induced acute lung injury via modulation of the MAPK/NF-κB signaling pathway in rats. Int. Immunopharmacol. 2020, 83, 106444. https://doi.org/10.1016/j.intimp.2020.106444
- Following sentence were deleted to the Material & methods part
By an Image-Pro® plus computer-aided image analysis system (Media Cyberbetics Inc., Silver Spring, MD, USA) attached to a light microscope (Olympus, Tokyo, Japan), images of picrosirius red-stained slides were captured.
Q4. The result part of the article is only a simple list of results. It is recommended that the results and discussion parts be written in detail and increased in depth.
Author answer: According to reviewer’s comment, we have revised the Result and Discussion parts to supplement the manuscript.
- Following sentence were added to the Results part
3.1. BALF cells, lung injury score, and pulmonary fibrosis
The BALF cell counts are shown in Figure 1B. The BALF cell count was decreased by PM10-induced lung fibrosis (P<0.05). But, pirfenidone administration enhanced the BALF cell count (P<0.05). The evaluation of lung histopathology and the extent of pulmonary fibrosis is pre-sented in Figure 1C. After 72 days of PM10 exposure, notable pathological changes such as intra-alveolar hemorrhage, interstitial edema, and infiltration of inflammatory cells were evident within the alveolar spaces. Moreover, fibrotic bands or nodular structures developed in the lung tissue of the PM10-injected group. These findings demonstrate that PM10 exposure significantly aggravated both lung injury and fibrosis. However, treatment with pirfenidone alleviated the formation of fibrotic bands and nodules, leading to a marked reduction in pulmonary fibrosis (P<0.05). Furthermore, pirfenidone effectively mitigated inflammatory infiltration, interstitial edema, and fibrotic structures, resulting in a significant improvement in lung injury scores (P<0.05).
3.2. Pro-inflammatory cytokines
The levels of TNF-α, IL-1β, and IL-6 expression, which are called pro-inflammatory cytokines in lung tissue, are presented in Figure 2. The levels of TNF-α, IL-1β, and IL-6 ex-pression were enhanced in PM10-induced pulmonary fibrosis (P<0.05). But, pirfenidone application inhibited the levels of TNF-α, IL-1β, and IL-6 expression (P<0.05). These findings suggest that pirfenidone effectively suppresses the inflammatory cascade triggered by PM10 and may play a protective role in mitigating inflammation-associated pulmonary fibrosis.
3.3. Fibrotic factors
The levels of TGF-β, hydroxyproline, and CTGF expression, which are named as fi-brotic factors, in lung tissues are presented in Figure 3. PM10 injection-induced lung fibro-sis enhanced the levels of TGF-β, hydroxyproline, and CTGF expression (P<0.05), but pirfenidone administration decreased the levels of TGF-β, hydroxyproline, and CTGF ex-pression (P<0.05). These results indicate that pirfenidone effectively suppresses fibrogenic signaling and collagen synthesis, suggesting its therapeutic potential in attenuating PM10-induced lung fibrosis.
3.4. TAK1/MKK3/p38 signaling pathway
Expression levels of TAK1/MKK3/p38 are presented in Figure 4. PM10 injec-tion-induced lung fibrosis enhanced TAK1/MKK3/p38 expression (P<0.05). Pirfenidone administration significantly decreased their expression levels (P<0.05), suggesting its role in suppressing this pro-fibrotic signaling pathway.
3.5. Collagen type I
Collagen type I expression detected by immunohistochemistry in the lung tissue is presented in Figure 5. PM10-induced pulmonary fibrosis enhanced collagen type I expres-sion in lung tissue. But, pirfenidone administration suppressed collagen type I expression, suggesting its anti-fibrotic effect on collagen deposition.
3.6. Fibronectin
Fibronectin expression detected by immunohistochemistry in the lung tissue is pre-sented in Figure 6. PM10-induced pulmonary fibrosis increased fibronectin expression in lung tissue. But, pirfenidone administration suppressed fibronectin expression, indicating its inhibitory effect on fibrotic remodeling.
- Following sentence were added to the Discussion part
In comparison with human lung pathologies associated with PM exposure, lung injury observed in a mouse model with PM₁₀ instillation closely resembles early-stage acute lung injury (ALI), characterized by alveolar inflammation and disruption of the epithelial barrier. When combined with fibrogenic factors, PM10 exposure may also induce fibrotic changes in the lung parenchyma, partially reflecting early features of pulmonary fibrosis in humans [24-26].
In the present study, intratracheal administration of PM induced alveolar capillary congestion, infiltration of inflammatory cells, and pulmonary fibrosis. These pathological changes led to a higher lung injury score compared to the non-PM-treated group.
Likewise, the present results demonstrated that the lung damage caused by PM administration may be due to be an increase in TNF-α, IL-1β, and IL-6, the pro-inflammatory cytokines, in lung tissue.
These results suggest that pirfenidone is therapeutically effective against PM-induced pulmonary fibrosis. However, lung function test including arterial blood gas analysis after PM exposure was not measured in this study. For this reason, further studies using various lung function tests are needed to improve PM-induced fibrosis.
Round 2
Reviewer 3 Report
Comments and Suggestions for Authors
Recommendations are as follows :
1. It is suggested that the author should elaborate the research background in the introduction, indicate the shortcomings of other therapeutic drugs or methods, and highlight the medical value of pirfenidone in modern medical applications;
2. There are few discussions on the mechanism of pirfenidone in the treatment of pulmonary fibrosis, no in-depth analysis, and lack of scientific effectiveness ;
3. The level of English needs to be improved;
4. There is still no detailed explanation of the reasons for the use of PM10 for pulmonary fibrosis modeling in the article;
5. In the abstract and introduction, it is suggested to add new experimental content.
Author Response
Answers to Reviewers’ Comments
Journal name: Biomedicines
Manuscript ID: biomedicines-3516725
Type of manuscript: Article
Title: Pirfenidone alleviates against fine particulate matter-induced pulmonary fibrosis modulating via TGF-β1/TAK1/MKK3/ p38MAPK signaling pathway in rats
Authors: Jun-Seok Sung, Il-Gyu Ko, Lakkyong Hwang, Sang Hoon Kim, Jin Hee Han, Jung Won Jeon, Sae Rom Kim, Jeong Mi Lee, Cheon Woong Choi *
We sincerely appreciate for your kind advice and comment to our manuscript. We revised to manuscript according to the reviewer’s comments. We added new experimental data, and modifications were expressed in red.
Q1. It is suggested that the author should elaborate the research background in the introduction, indicate the shortcomings of other therapeutic drugs or methods, and highlight the medical value of pirfenidone in modern medical applications.
Author answer: First all, thank you for your kind advice and comments. According to reviewer’s comment, we added new sentences were inserted in manuscript.
- Following sentence were added to the Introduction part
As small PM initiates or worsens the symptoms of numerous lung diseases, it presents a growing public health threat [3].
Despite advances in understanding pulmonary fibrosis, current treatments like corticosteroids and immune-suppressants offer limited efficacy and often cause side effects. As they primarily address inflammation rather than fibrosis itself, there remains a pressing need for safer, more targeted anti-fibrotic therapies.
However, its potential to mitigate PM-induced pulmonary fibrosis remains poorly un-derstood. Clarifying this could expand pirfenidone’s therapeutic utility and provide novel insights into air pollution-related respiratory diseases.
- Following sentence were deleted to the Introduction part
But, it is unclear whether pirfenidone is effective in treating pulmonary fibrosis due to PM.
Q2. There are few discussions on the mechanism of pirfenidone in the treatment of pulmonary fibrosis, no in-depth analysis, and lack of scientific effectiveness.
Author answer: Pirfenidone is an antifibrotic and anti-inflammatory agent used in idiopathic pulmonary fibrosis (IPF). To date, the exact mechanism has not yet been fully elucidated. The results of this study only suggest various possibilities for the therapeutic efficacy of pirfenidone. It is very unfortunate that this study did not elucidate the mechanism in more depth. Therefore, further detailed and additional mechanistic studies are needed in the future. Based on the reviewer's comments, the content of the discussion section has been revised and the validity section has been supplemented.
- Following sentence were changed to the Discussion part
Pirfenidone, an anti-fibrotic agent, has been shown to suppress the production of pro-inflammatory cytokines and chemokines, thereby reducing inflammatory responses and T-cell activity [27]. Additionally, pirfenidone inhibits the expression of intracellular adhesion molecule-1 and promotes the expression of anti-inflammatory cytokines [28-29]. In the current study, pirfenidone significantly reduced the production of pro-inflammatory cytokines in the lungs, further confirming its anti-inflammatory properties.
PM exposure leads to marked inflammation, which over time results in pulmonary fibrosis and lung damage [30-31]. The previous study demonstrated that fibrosis was progressively aggravated in mice exposed to PM for 8 weeks, suggesting that PM exposure directly promotes fibrosis [31]. However, the precise mechanisms underlying PM-induced lung fibrosis remain unclear. One potential pathway is the TAK1/MKK3/p38 signaling cascade, which regulates TGF-β, a major mediator of fibrosis [32-33].
TAK1 has been shown to play a critical role in the activation of TGF-β-induced MKK3 and subsequent collagen production in mesodermal cells [29]. Additionally, the expression of TGF-β-induced fibronectin in fibroblasts is mediated by TAK1, and TAK1-deficient fibro-blasts exhibit a reduced profibrotic response to TGF-β stimulation [7,32-33]. Notably, the profibrotic function of TAK1 signaling has been observed in lungs exposed to PM, where TAK1 activation leads to increased phosphorylation of p38 MAPK, promoting fibrotic events in lung tissue [32-33].
Q3. The level of English needs to be improved.
Author answer: According to editor point out, we carefully checked spelling and grammar. We have made efforts to improve the English version of this manuscript.
Q4. There is still no detailed explanation of the reasons for the use of PM10 for pulmonary fibrosis modeling in the article.
Author answer: In this study, PM10 was used to model pulmonary fibrosis due to its well-established role in air pollution-related respiratory diseases. PM10, defined as particulate matter with an aerodynamic diameter of ≤10 µm, is a major component of air pollution, originating from sources such as industrial emissions, vehicular exhaust, construction activities, and natural dust (Pope and Dockery, 2006; Brook et al., 2010). Due to their small size, PM10 particles can bypass the upper respiratory defenses and deposit deep within the alveolar spaces, where they interact with epithelial cells and alveolar macrophages, initiating inflammatory and fibrotic responses (Kim et al., 2015). Chronic PM10 exposure has been linked to increased respiratory morbidity, exacerbation of chronic obstructive pulmonary disease, and progression of interstitial lung diseases including idiopathic pulmonary fibrosis (Johannson et al., 2018; Winterbottom et al., 2018). These findings establish PM10 as a relevant agent for modeling environmentally triggered fibrotic lung disease and for evaluating therapeutic interventions such as pirfenidone. According to reviewer’s comment, we added new sentences were inserted in manuscript.
[Reference]
Pope CA3rd, Dockery DW. Health effects of fine particulate air pollution: lines that connect. J Air Waste Manag Assoc. 2006;56(6):709-42.
Brook RD, Rajagopalan S, Pope CA3rd, Brook JR, Bhatnagar A, Diez-Roux AV, Holguin F, Hong Y, Luepker RV, Mittleman MA, Peters A, Siscovick D, Smith SC Jr, Whitsel L, Kaufman JD; American Heart Association Council on Epidemiology and Prevention, Council on the Kidney in Cardiovascular Disease, and Council on Nutrition, Physical Activity and Metabolism. Particulate matter air pollution and cardiovascular disease: An update to the scientific statement from the American Heart Association. Circulation. 2010;121(21):2331-78.
Kim KH, Kabir E, Kabir S. A review on the human health impact of airborne particulate matter. Environ Int. 2015;74:136-43. doi: 10.1016/j.envint.2014.10.005
Johannson KA, Vittinghoff E, Morisset J, Wolters PJ, Noth EM, Balmes JR, Collard HR. Air Pollution Exposure Is Associated With Lower Lung Function, but Not Changes in Lung Function, in Patients With Idiopathic Pulmonary Fibrosis. Chest. 2018;154(1):119-125.
Winterbottom CJ, Shah RJ, Patterson KC, Kreider ME, Panettieri RA Jr, Rivera-Lebron B, Miller WT, Litzky LA, Penning TM, Heinlen K, Jackson T, Localio AR, Christie JD. Exposure to Ambient Particulate Matter Is Associated With Accelerated Functional Decline in Idiopathic Pulmonary Fibrosis. Chest. 2018;153(5):1221-1228.
- Following sentence were added to the Discussion part
These findings establish PM10 as a relevant agent for modeling environmentally triggered fibrotic lung disease and for evaluating therapeutic interventions such as pirfenidone. PM10 was used in this study to model pulmonary fibrosis because it can penetrate deep into the lungs and trigger oxidative stress, inflammation, and fibrotic responses.
Q5. In the abstract and introduction, it is suggested to add new experimental content.
Author answer: According to reviewer’s comment, we added new sentences were inserted in abstract and introduction.
- Following sentence were added to the Abstract part
PM administration increased the lung injury score, level of lung fibrosis, and production of pro-inflammatory cytokines.
- Following sentence were added to the Introduction part
In the present experiment, bronchoalveolar lavage fluid (BALF) examination was done and the level of lung fibrosis was measured using picrosirius red staining. Lung in-jury scores were calculated using hematoxylin and eosin (H&E) staining.